# Data Analysis Model Design of Health Service Monitoring System for China’s Elderly Population: The Proposal of the F-W Model Based on the Collaborative Governance Theory of Healthy Aging

**DOI:** 10.3390/healthcare9010009

**Published:** 2020-12-23

**Authors:** Liping Fu, Tao Teng, Yuhui Wang, Lanping He

**Affiliations:** 1College of Management and Economics, Tianjin University, Tianjin 300072, China; lpf3688@126.com (L.F.); tengtao.321@163.com (T.T.); lanping@tju.edu.cn (L.H.); 2Center for Social Science Survey and Data, Tianjin University, Tianjin 300072, China

**Keywords:** data analysis model design, health service monitoring system, F-W model, cooperative governance, healthy aging

## Abstract

In the era of artificial intelligence, big data and 5G, health care for elderly people is facing an important digital transformation. The objective of this study is to design the data analysis module of the elderly health service monitoring system (HSMS) and attempt to put forward a new healthy aging (HA) model that is applicable not only to the individual HA, but also to the regional HA system. Based on the HA theory of collaborative governance, we divided the elderly HSMS into four modules, including physical health, mental health, ability of daily activity, and social participation. Then, factors that influence HA were assessed by stepwise logistic regression to build the analysis model, using the public micro-panel data of the China Health and Retirement Longitudinal Survey (CHARLS). Age (odds ratio (OR) = 1.55 (95% confidence interval (CI): 1.06–2.27)), living in urban areas (OR = 1.57 (95% CI: 1.03–2.39)), being literate (OR = 1.51 (95% CI: 1.01–2.23)), expecting to get long-term health care in the future from their grown children (OR = 1.69 (95% CI: 1.10–2.61)) and having literate grown children (OR = 2.01 (95% CI: 0.26–0.97)) had a significant positive impact on HA of elderly people. Therefore, the F-W (factors and weighs, also family and welfare) model is proposed in this paper. The outcomes can contribute with designing HSMS for different provinces and several different regions in China and leave a door open to improve the model and algorithm application for HSMS in the future studies.

## 1. Introduction

The 21st century is not only the era when people are witnessing the aging of the population but also an important moment of digital transformation, giving room for improvements and innovations of artificial intelligence, big data and 5G all across the world. China has become the country with the largest elderly population and the fastest aging growth rate over the past 20 years [1,2]. At the same time, the country’s economic development and technological changes are also rapidly upgrading. Consequently, to use the latest technology to deal with the challenge of health quality among elderly populations has become one of the most important research issues [3,4,5].

In the field of medical big data, data-driven computational methods are increasingly needed to quickly and accurately analyze large-scale biological data, therefore, technology of health service monitoring system (HSMS) is developing rapidly [6,7]. According to the prediction of Internet Data Center (IDC), the global data volume is doubling every year. In 2011, the total volume created and copied was 1.8zb and it is estimated that this number will reach 35 zb or more by 2020, with individuals being responsible for 75%. The use of wearable health equipment and the standardization of electronic medical records is expected to generate more than 605 tb of data per person per day in terms of physical health, mental health, daily activities, healthy environment, social participation and other aspects [8,9]. The rapid generation of such unpredictable health care big data is a huge challenge to information infrastructure construction. With the accumulation of data and technological progress, the design and development of HSMS is a topic that has been attracting a lot of attention [10,11,12].

The elderly health service detection system can be divided into two situations. The first one regards monitoring for elderly individuals or a group, and the second is more focused on certain regions (e.g., cities or provinces). Specifically, HSMS can be divided into nine levels: individual, family, community, cluster, township, city, province, country and global. There can be large and small groups within clusters, communities, and cities, including elderly groups placed in medical care homes [13]. However, the divisions may not be clear-cut since, for instance, elderly empty nesters exist in both rural and urban areas [14].

Currently, China and all the nations seem to not have a HSMS for the elderly covering all groups and regions. It is mainly limited by technical, economic and medical service accessibility. Considering that technology has been developing faster, population-centered big data tend to continue to be established. Therefore, the collection, transmission and storage of data might no longer be a problem, shining a light on questions like how to analyze the data. That said, how to build the data analysis model of the elderly health service system is one of the key research tasks in the field of health care at present and in the future [15,16,17]. Even if different data are applied, the model framework of the elderly HSMS is interlinked. Using existing survey data (e.g., CHARLS) is an option before building big data on healthcare [18,19].

In addition to understanding data analysis, constructing a subsystem for health monitoring is also equally important and this is also the first major innovation and marginal contribution of this study.

The World Health Organization first put the concept of healthy aging forward in 1999. Healthy aging is defined as the process of developing and maintaining the healthy life of the elderly. Functional development refers to the health-related factors that individuals can live and act according to their own ideas and preferences [20,21]. The environment includes all the external factors that make up the individual life of the elderly, which can be divided into three levels: individual, family and society from micro to macro. The concept has been widely used in academic research and government policy fields since it was proposed. However, there has been a lack of consensus on its composition, definition and measurement methods [22,23]. Therefore, the process of integration with science and technology and the application of specific digital technology are still not completely deciphered [24,25,26].

Taking into consideration the perspective of global public health development, the technical application conditions and opportunity of healthy aging are mature, mainly manifested in the current global prediction and control of chronic diseases and the extension of the world’s per capita health life expectancy. From 2000 to 2016, the global overall life expectancy increased by more than 8%, and the average healthy life expectancy increased from 59 to 63 years [27].

China’s overall life and health life expectancy has also increased from 72 and 65 years in 2000 to 76.4 and 68.7 years, respectively. In 2016, China put forward the healthy China 2030 strategy and healthy aging strategy implementation action, which further supplemented the connotation of this issue. The main connotation of healthy aging in China is health-centered, which promotes the daily activities and social participation ability of the elderly. The main goal is to further reduce the incidence rate of chronic diseases and improve the rates of longevity [28,29,30].

In addition to the module content design of HSMS, how to carry out specific application of HSMS remains a problem. At present, China has put forward a lot of policy support in promoting healthy aging, medical big data, and the elderly HSMS. However, there is a lack of coordination between micro and macro, and health policies might not fit families and the aging society well [31,32].

In China, under the influence of traditional culture, the families of the elderly not only include their own families, but also are influenced by the new members brought by their grown children when they get married. Children’s care and family support have an important impact on the physical and mental health of the elderly [33]. In this case, considering the perspective of collaborative governance, distinguishing and testing the mode of family pension, community pension and institutional pension can also be considered a possible marginal contribution of this study [34,35]. At present, 90% of the elderly population in China chooses the home-based care model. It is necessary and representative to screen the data and research objects [36,37].

Moreover, it is still necessary to further build a healthy city and an elderly friendly city. In China, the regional economy and health care level are quite different. Based on the theory of cooperative governance, it is necessary to measure the level of healthy aging in each region and construct appropriate HSMS that are applicable to each case. Therefore, the third innovation point and possible marginal contribution of this study is to construct individual micro and macro HSMS by measuring and testing individual micro influencing factors and the weight of health aging [38,39,40].

The variables related to “the second family of the elderly” mainly include the physical and mental health, marital status, education and the economic situation of their grown children [41,42,43,44,45]. In order to avoid the statistical fallacies caused by the interaction between indicators, this study first tests the indicators of “the first family” and “the second family” separately, and then tests all variables together by stepwise logistic regression.

## 2. Materials and Method

### 2.1. Materials

The analysis was based on secondary data collected as part of the China Health And Retirement Longitudinal Study (CHARLS) 2015. The CHARLS survey began officially in 2011, and was conducted every two years [46]. The data from it have been widely used in recent years. In each module (Demographic backgrounds, Family, etc.), some of the questions specifically ensure the authenticity and accuracy of the data, such as “How often did the respondent receive assistance in answering the section Demographics”? (question number BF008 in the CHARLS questionnaire). CHARLS is publicly available on the website: http://charls.pku.edu.cn/.

First, for the dependent variables, as presented in Table 1 and Table A1, “Good physical health” (Y1) was defined as no disability and fewer chronic diseases (no more than two). The chronic diseases included hypertension and hyperglycemia.

The “Good ability of daily activities” (Y2) scale of CHARLS 2015 was basically consistent with the common ADL (Activities of Daily Living) scale. Ten main daily activities were considered in this operational definition, including dressing, bathing, eating, getting into or out of bed, using the toilet, controlling urination and defecation, doing household chores, shopping for groceries, making phone calls and taking medications. Choices for each item were graded “1 = I don’t have any difficulty”, “2 = I have difficulty but can still do it”, “3 = I have difficulty and need help” and “4 = I cannot do it”. Grade 1 or 2 for each activity indicates that the older person is independent in this daily activity.

For each item, a grade less than or equal to 2 was considered as meeting the criterion. The Cronbach’s α of scale was 0.784 in this study, and if the item of making phone calls was removed, the Cronbach’s α of scale would be 0.801. However, considering the important role of the telephone in daily life and the fact that all of the respondents had telephones in their homes, we kept this item in this study.

The “Good mental health” (Y3) scale of CHARLS 2015 was similar to the common HAD (Hospital Anxiety and Depression) scale. The Cronbach’s α of scale was 0.802 in this study. Ten scenarios were simulated and four choices for each one were graded: “1 = Rarely or none of the time”, “2 = Some or a little of the time”, “3 = Occasionally or a moderate amount of the time” and “4 = Most or all of the time”. A total score of less than or equal to 20 was considered as good mental health.

Then, for the last criterion of “Good social participation” (Y4), the respondents were asked to report the frequency of their social activity during the last few months. Eight social activities were considered in this operational definition, including voluntary or charity work, caring for a sick or disabled adult, providing help to family, friends or neighbors, attending an educational or training course, interacting with friends, going to a sport, social or other kind of club, taking part in a community-related organization. There were three frequency choices for the respondents, including almost daily, almost every week and not regularly. The number and frequency of social activities were used to measure whether the respondents were active. It was considered that the standard had been met when the number of activities attended was no less than two in the last month.

Interviewees which reached the criteria of “Good physical health”, “Good ability of daily activities”, “Good psychological health” and “Good social participation” were considered to be healthy aging (Y).

Based on question number Bb000 in CHARLS 2015, the sample data with a total of 795 respondents were collected. We found that 68.6% of the respondents reached the standard of “no disability and less chronic diseases”, less than half (39.4%) of the respondents were active participants in society, most (84.7%) had good ability to perform daily activities, 61.4% were in a good mental state, and 17.2% met the criteria of healthy aging.

Then, for the independent variables, as presented in Table 2, “Gender” (X1), “Age” (X2), “Residence” (X3), “Educational status ” (X4), “Marital status” (X5), “Expectation of Long-term Care in the future from grown children” (X6), “Educational status of grown children” (X7), “Living place of grown children” (X8), “Marital status of grown children” (X9), “Physical condition of grown children” (X10), “Housing Property status of grown children” (X11) and “Elderly parents provide Inter-generational care for grown children’s babies” (X12) were included in this study and they could be divided into two parts of the first family (from X1 to X6) and the second family (from X7 to X12).

Choices and grades were reduced to a discrete form, such as “male or female”, “literate or illiterate”, or “yes or no”. Among the respondents, 447 were female and 348 were male. There were 307 young people aged between 50 and 60 years old, and 488 people over 60 years old. 177 people lived in cities and 618 people lived in rural areas. 397 respondents were literate and 417 respondents were illiterate. 602 respondents were married and 193 respondents were unmarried. There were 528 elderly people who expected to receive long-term care from their grown children, and 267 were not expected to receive long-term care. 617 respondents’ grown children were educated and 312 respondents were living with adult children. There were 624 respondents’ grown children married. 467 respondents’ grown children were in good health and 409 had at least one property. 400 respondents provided intergenerational care for their grown children’s babies.

The respondents who were male, younger, literate, married, lived in cities and whose grown children were educated, married, healthy, not living with parents, or had at least one house made up a higher proportion of the HA population.

### 2.2. Logistic Regression Analysis of the Indicators for the HA

Table 3 shows that the grouping regression results are basically consistent with the correlation analysis. The respondents who were younger (OR = 1.64 (95% CI: 1.10–2.45)), living in urban areas (OR = 1.66 (95% CI: 1.08–2.55)), educated (OR = 1.54 (95% CI: 1.02–2.34)), or expected to get health care in the future (OR = 1.70 (95% CI: 1.11–2.62)) were more likely to achieve healthy aging. It was also beneficial if their adult children were educated (OR = 2.48 (95% CI: 1.29–4.76)) and in good physical health (OR = 1.27 (95% CI: 1.05–1.55)).

As shown in Table 4, the variables with significant influences in all groups were gathered to control regression step by step. Putting all the variables together avoids the regression error caused by multicollinearity. Stepwise logistic regression was used to get the last significant indicators associated with HA.

In the first step, the influence of the children’s health status was no longer significant, and was removed. Then, the other significant indicators were removed one by one. In the last step, the respondents who were younger (OR = 1.55 (95% CI: 1.06–2.27)), living in urban areas (OR = 1.57 (95% CI: 1.03–2.39)), literate (OR = 1.51 (95% CI: 1.01–2.23)), expected to get long-term health care in the future (OR = 1.69 (95% CI: 1.10–2.61)), or had educated children (OR = 2.01 (95% CI: 0.26–0.97)) were more likely to achieve healthy aging. The detailed steps are shown in Table A2, Table A3, Table A4, Table A5, Table A6, Table A7, Table A8 and Table A9 in the Appendix A.

In order to find out whether the amendment of the four dimensional model of HA in this study is reasonable, as an auxiliary analysis, the main indicators to the four-dimensional criteria were tested. As shown in Table 5, the physical health status of children is a protective factor for the criterion of “no disability and few chronic diseases”, residence of the elderly is a protective factor for the criterion of “active social participation”, the age and educational level of the elderly and their children’s educational status are protective factors for the criterion of “good abilities of daily activities”. However, educational status, expectations for long-term health care, and children’s physical health status of the respondents had significant impacts on the criterion of “good mental health”. Comprehensive impact statistics can be found in Appendix A
Table A10.

## 3. Regional Healthy Service Monitoring System Design

To summarize the above analysis, the healthy aging of middle-aged and elderly people in China is mainly influenced by age, educational status, residence, and expectation for long-term health care in the future. This result can be applied to the design of an individual or group elderly health system. According to the literature and China’s social reality, we assumed that healthy aging has certain regional characteristics. Different regions have different characteristics of healthy aging, and macro regional healthy aging is mainly affected by the micro elderly population.

As presented in Table 6, if we want to build urban, provincial and even national HSMS, the corresponding criteria of HA in NSD (National Statistical Database) will be the regional aging level, educational level, urbanization level and level of family care. Finding these criteria is the main function of assessing indicators associated with HA.

In the process of logical regression, *β* is equal to the natural logarithm of the OR (odds ratio). The results of the previous logistic regression indicated that X2, X3, X4, X6 and X7 had a significant impact:(1)Y=β2X2+β3X3+β4X4+β6X6+β7X7

Therefore, we could construct a scoring index statistical model of *HA* as follows:(2)$$HA=\overset{n}{\underset{i=1}{\sum }}F_{i}W_{i}$$
where *HA* stands for the scoring index of the proportion of healthy aging, *W* stands for the weight and the logistic regression coefficient (*β*) will be the weight coefficient in this study. *F* represents the factors that had significant effects on regional healthy aging. We define this model as the F-W model.

The education level of the region is measured by the ratio of the average education time of the regional population. The level of urbanization is measured by the ratio of urbanization rate to the national average. The level of family care is measured by the ratio of the number of households to the national average. The regional age level is measured by the ratio of the ratio of the reciprocal value of the old-age dependency ratio to the national average.

The scoring index of healthy aging in various regions of China is estimated by selecting the data of 30 provinces from 2005–2015 statistical yearbooks. It does not include the province of Hong Kong, Macao, Taiwan and Xizang province due to the lack of data.

There are significant differences in the degree of healthy aging in 2005, 2010 and 2015 in various provinces (Figure 1). Over time, little change occurred in the regional differences in China from 2005 to 2015.

The average healthy aging index in the provinces from 2005 to 2015 continues to be calculated and ranked from higher to lower (as shown on the horizontal axis of Figure 2). It can be seen that there are two main kinds of regions with a high rate of healthy aging, showing the obvious polarization phenomenon. The first group includes Beijing, Tianjin, Shanghai, and Guangdong provinces, which are economically developed with a large population and a high degree of aging. The second group includes Ningxia, Qinghai, Hainan and Xinjiang provinces, which are less economically developed with population outflow. Then, based on the rise and fall of the healthy aging level in each province from 2005 to 2010 and 2010 to 2015, the provinces can be grouped into four areas: region 1 (HA score goes up first from 2005 to 2010 and then down from 2010 to 2015), region 2 (HA score has been falling from 2005 to 2015), region 3 (HA score goes down first from 2005 to 2010 and then up from 2010 to 2015) and region 4 (HA score keeps going up from 2005 to 2010) in Table 7.

Table 8 shows that the average index of healthy aging in the north of China, especially in the Northeast, North and East China, is descending, while those in the south, especially in the Pearl River Delta, Southeast Coast Area and the Southwest are ascending. Comparing and further discussing the differences of regional agglomeration, based on the economic and social development, as well as the distribution of urban agglomerations and metropolitan areas, China is divided into eight regions, such as the Yangtze River Delta, the Pearl River Delta, North China, and Northeast China. It can be seen that China’s healthy aging level has obvious differences among various provinces and regions. The Beijing-Tianjin-Hebei (112.49), Pearl River Delta (104.56) and Yangtze River Delta (104.43) regions have the highest HA indexes.

## 4. Discussion

Following an operational four-dimensional model, this study initially aimed to assess the prevalence and indicators of HA among Chinese home-dwelling elderly people. The results indicated that less than a fifth (17.2%) of elderly people reached HA. However, because of the differences in selection criteria, sample data and independent variables, it is inappropriate to make a simple comparison with other studies. Compared with previous studies, the proportion of HA among Chinese elderly people is still not high in general. This shows that the healthy aging process of China’s elderly population still deals with some challenges [47,48].

The family planning policy has affected the process of population aging with human intervention and China has to handle the issue of a population aging before getting wealthy enough. The influx of young children into big cities and the lack of family support function has the potential to negatively affect the healthy aging process of China’s elderly population [49].

Initiatives to support for the elderly, change of intergenerational relationships and regional equilibrium constitute three vital indicators that will shape China’s sustainable development of public health in the future [3,4,9,16]. Therefore, now is the moment that the authorities should invest in ways to improve the health quality of the aging population and increase the investment in health care [1,2,5].

Then, through the theoretical framework of HA from the perspective of family structure change, this study assessed factors of HA among Chinese home-dwelling elderly people by stepwise logistic regression. Age, living in urban areas, being literate, and “expecting to get the long-term health care in the future” had a significant positive impact on HA, while showing no difference in gender and marital status.

The conclusions of testing age and education status are in accordance with the finding of some previous studies [44,45,46,47]. Generally, the elderly who are expected to receive long-term care in the future always have good family support. The reason why no gender difference was found is probably because the traditional Chinese gender concept of “men being superior” has changed, and women now have more means to achieve higher social status. Most of the elderly who choose home-based care are couples with a stable marital status. Meanwhile, the educational status of grown children was the only significant influencing factor of HA in “the second family”; marital status, living place, housing property status and intergenerational care did not show a significant impact.

The HA situation in various regions of China, the trend of change in the past decade, and regional differences is clear through charts. China’s aging process can be divided into two major time stages—a period from 2005 to 2010 and another one from 2010 to 2015. According to the changing trend (rising and falling) of the HA index scores in both stages, HA areas in China could be divided into four regions, including region 1 (first rising and then falling HA score), region 2 (continuously decreasing HA score), region 3 (first falling and then rising HA score), and region 4 (continuously rising HA score). The scoring index of HA in each region was counted.

The findings indicated the provincial and regional differences in HA in China. The HA level of the eight traditional regions was measured, including important areas like the Beijing-Tianjin-Hebei region (112.49, baseline 100), the Pearl River Delta region (104.56) and the Yangtze River Delta region (104.43), that had the highest HA scoring index. In general, regions with better economic conditions provide quality health systems and medical resources, which are the preconditions for promoting HA [48]. In different areas, the design of HSMS also has corresponding differences, to avoid the blind application of digital technology from causing waste of health care resources.

At the same time, some limitations must be acknowledged. First, the results were carried out according to our four-dimensional model, and some new indicators or different evaluation systems might be applicable in the future. That said, a statistical construction of an indicator system for HA should be further explored. Second, in the specific implementation process, the change of questionnaire design and weighting of each criterion were also impactful. The assignment could be more detailed and variables like marital status, for instance, could be further divided into married, divorced, widowed, never married and other situations to further explore the impact of different marital statuses on the same scenario. Third, as an important variable, educational status might require a better shape and focus in future studies. The educational level of the elderly, the educational level of their grown children, and the re-education and re-employment situation of the elderly populations are worth further determination. There are more possibilities for the matching of personal and regional HSMS. Furthermore, some social variables such as economic status and government can be developed in different ways, with more prospects. In general, most variables among the elderly people have been included in this study and the estimated result is applicable.

## 5. Conclusions

From a global perspective, significant improvements in life expectancy and continuous population aging have become a pervasive phenomenon in nearly all countries and regions, regardless of whether they are developed or still developing [46,47,48]. The remarkable increase of health needs an impetus source of social progress, but it also brings huge challenges to health-related quality of life among elderly populations and public health services. The aging problem and HA have attracted great attention from the international academic community.

Through the theoretical framework of HA first, this study provides an extension concept of HA, including good physical health, good mental health, active social participation and good ability to perform daily activities. Moreover, this paper innovatively combines the micro individual HA with the macro regional HA.

The micro-perspective transformation includes the perspective of family structure change. The development of HA among Chinese elderly people should be based on the support of family, considering the traditional culture and the current social reality of China. Therefore, it is necessary to research HA from the points of view of a possible change in the family structure. Taking family structure as the core variable, it is possible to divide the family of the elderly into two parts, including “the first family” with spouse and “the second family” of their children. This is why the F-W (factors and weighs, also family and welfare) model is proposed in this paper.

From the macro perspective, the regional difference was analyzed based on provincial data from 2005 to 2015. The outcomes make it possible to design a fit HSMS for the elderly and another one that can function in different provinces and several different regions in China. On this basis, it is possible to keep using new and existing dates to improve the algorithm application in new ways.

In the future research, F-W model could have better scalability. It is possible to use research findings to also develop large-scale HSMS applied to communities, towns, cities, provinces, countries and other nations. Faced with the rapid development of digital technology, the integration of elderly health, public health, health policy and health technologies is more and more in-depth. It is predicted that the application of the health care industry, intelligent medical care and digital technology will provide more and more possibilities for elderly health.

## Figures and Tables

**Figure 1 healthcare-09-00009-f001:**
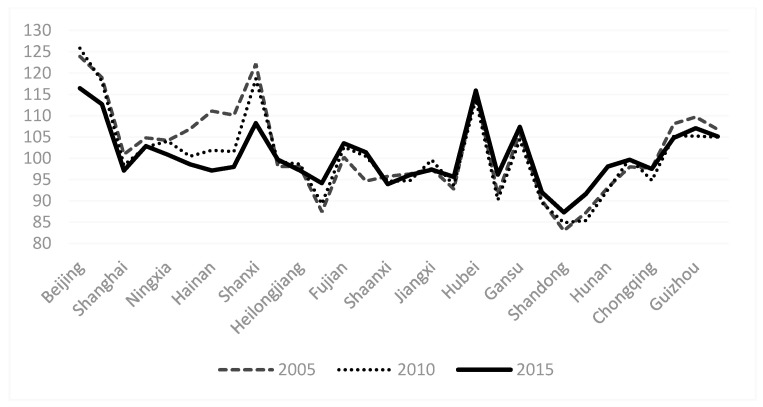
Index of HA level in 30 provinces of China in 2005, 2010 and 2015.

**Figure 2 healthcare-09-00009-f002:**
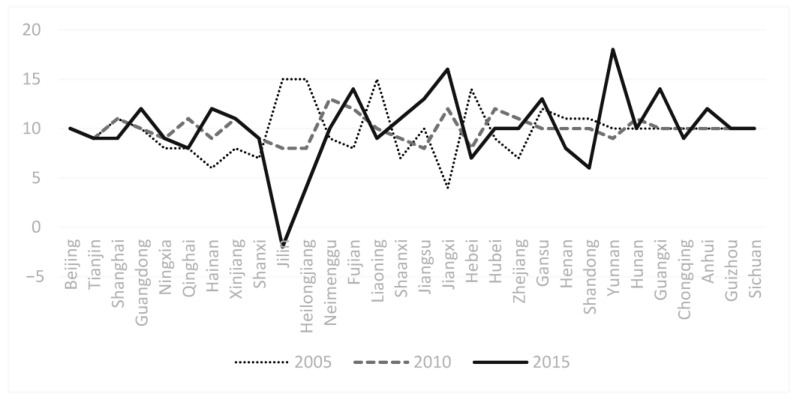
Ranking of average HA index from 2005 to 2015 and Trends of Provinces.

**Table 1 healthcare-09-00009-t001:** Descriptive statistics of dependent variables.

Characteristics	Specific Indicators	N	%
Good physical health(Y1)	No disability and fewer chronic diseases	545	68.6%
Good ability of daily activities (Y2)	Grade 1 or 2 for each activity	313	39.4%
Good mental health (Y3)	score of less than or equal to 20	673	84.7%
Good social participation (Y4)	the number of activities attended is no less than two in the last month	488	61.4%
Healthy aging (Y)	met the four standards of Y1–Y4	137	17.2%

Note: N, number of older adults meeting certain criteria; %, corresponding percentage.

**Table 2 healthcare-09-00009-t002:** Descriptive statistics of independent variables.

Variables	Specific Indicators	N	N (%)	N (%)	χ^2^	p
			Non-HA	HA		
Gender (X1)	Male	348	282 (42.8%)	66 (48.1%)	1.30	0.25
	Female	447	376 (57.2%)	71 (51.8%)		
Age (X2)	younger(50–60)	307	239 (36.3%)	68 (49.6%)	8.47	0.004
	older(60–90)	488	419 (63.7%)	69 (50.4%)		
Residence (X3)	Urban	177	134 (20.3%)	43 (31.3%)	7.95	0.005
	Village	618	524 (79.7%)	94 (68.7%)		
Educational status (X4)	Literate	397	296 (45%)	83 (60.6%)	11.06	0.001
	Illiterate	417	362 (55%)	54 (39.4%)		
Marital status (X5)	Married	602	492 (74.7%)	110 (80.2%)	1.87	0.17
	Unmarried	193	166 (25.3%)	27 (19.7%)		
Expectation of Long-term Care in the future from grown children (X6)	Yes	528	424 (64.4%)	104 (75.9%)	6.69	0.01
	No	267	234 (35.6%)	33 (24.1%)		
Educational status of grown children (X7)	Literate	617	535 (81.3%)	126 (91.9%)	9.200	0.002
	Illiterate	134	123 (18.6%)	11 (8%)		
Living place of grown children (X8)	with parents	312	261 (39.6%)	51 (37.2%)	0.28	0.59
	Not with parents	443	367 (60.4%)	86 (62.8%)		
Marital status of grown children (X9)	Married	624	516 (78.4%)	108 (78.8%)	0.01	0.91
	Unmarried	171	142 (21.6%)	29 (21.2%)		
Physical condition of grown children (X10)	Good	467	379 (57.5%)	88 (64.2%)	10.68	0.03
	Fair	328	279 (42.5%)	49 (35.8%)		
Housing Property status of grown children (X11)	Own at least a house	409	338 (51.3%)	71 (51.8%)	0.009	0.92
	No house	386	320 (48.7%)	66 (48.2%)		
Elderly parents provide Inter-generational care for grown children’s babies (X12)	Yes	400	334 (50.8%)	66 (48.1%)	0.30	0.58
	No or not yet	395	324 (49.2%)	71 (51.8%)		

Note: N, number of respondents; %, percentage; χ^2^, Chi-square test.

**Table 3 healthcare-09-00009-t003:** Grouping logistic regression of the indicators to HA.

Variables of the First Family	*β*	OR	95% CI	Variables of the Second Family	*β*	OR	95%CI
Gender (ref. female)	0.16	1.17	(0.78, 1.77)	Educational status of grown children (ref. illiterate)	0.91 **	2.48 **	(1.29, 4.76)
male				literate			
Age (ref. 60–90)	0.49 *	1.64 *	(1.10, 2.45)	Living place of grown children (ref. not at home)	−0.09	0.91	(0.61, 1.36)
40–60				live with respondents			
Residence (ref. village)	0.51 *	1.66 *	(1.08, 2.55)	Marital status of grown children (ref. unmarried)	−0.05	0.95	(0.58, 1.56)
urban				married			
Educational status (ref. illiterate)	0.43 *	1.54 *	(1.02, 2.34)	Physical condition of grown children (ref. not good)	0.24 *	1.27 *	(1.05, 1.55)
literate				good			
Marital status (ref. unmarried)	0.05	1.05	(0.64, 1.74)	Housing status of grown children (ref. do not)	0.02	1.02	(0.68, 1.54)
married				Own a house			
Expectations of long-term health care from grown children (ref. no)	0.53 *	1.70 *	(1.11, 2.62)	Elderly parents provide inter-generational care for grown children’s babies (ref. no)	−0.14	0.87	(0.60, 1.27)
yes				yes			

Note: β, Beta coefficient; OR, odds ratio; CI = Confidence Interval for odds ratio; *, *p* < 0.05; **, *p* < 0.01.

**Table 4 healthcare-09-00009-t004:** Stepwise logistic regression of the indicators to HA.

Y(OR/Std. Err.)	First Step	Second Step	Third Step	Forth Step	Fifth Step	Sixth Step	Seventh Step	Last Step
X1	1.228	1.236	1.239	1.239	1.232			
	(0.260)	(0.258)	(0.258)	(0.258)	(0.256)			
X2	1.697 *	1.713 *	1.699 *	1.705 *	1.682 *	1.631 *	1.487 *	1.554 *
	(0.383)	(0.377)	(0.366)	(0.367)	(0.361)	(0.346)	(0.91)	(0.300)
X3	1.606 *	1.602 *	1.604 *	1.604 *	1.601 *	1.556 *	1.543 *	1.573 *
	(0.351)	(0.349)	(0.349)	(0.349)	(0.349)	(0.335)	(2.01)	(0.338)
X4	0.727	0.725	0.725	0.726	0.723	0.671 *	0.681	0.663 *
	(1.158)	(0.156)	(0.156)	(0.156)	(0.156)	(0.136)	(−1.90)	(0.133)
X5	1.046							
	(0.273)							
X6	1.649 *	1.647 *	1.648 *	1.653 *	1.652 *	1.643 *	1.657 *	1.697 *
	(0.364)	(0.364)	(0.364)	(0.364)	(0.364)	(0.362)	(2.30)	(0.372)
X7	0.515	0.514	0.512	0.517	0.508 *	0.516	0.511 *	0.497 *
	(0.177)	(0.178)	(0.175)	(0.177)	(0.174)	(0.176)	(0.174)	(2.088)
X8	0.879	0.877	0.871	0.862				
	(0.185)	(0.182)	(0.171)	(0.178)				
X9	1.058	1.057						
	(0.282)	(0.282)						
X10	1.177	1.178	1.179	1.179	1.173	1.168	1.176	
	(0.124)	(0.124)	(0.124)	(0.124)	(0.123)	(0.122)	(0.122)	
X11	1.226	1.221	1.238	1.234	1.274	1.268		
	(0.275)	(0.272)	(0.265)	(0.256)	(0.267)	(0.266)		
X12	0.889	0.890	0.890					
	(0.172)	(0.172)	(0.172)					
_cons	2.769	2.791	3.013	2.832	2.646	3.203	3.745	5.21
	(0.145)	(0.146)	(1.589)	(1.463)	(1.340)	(1.501)	(1.688)	(0.000)

Note: OR, odds ratio; *, *p* < 0.05; _cons, constant.

**Table 5 healthcare-09-00009-t005:** Logistic regression of the main indicators to the four criteria of HA.

Variables	No Disability and Fewer Chronic Diseases	Good Social Participation	Good Ability of Daily Activities	Good Mental Health
Age (X2)	1.33	1.24	2.13 **	0.78
Residence (X3)	0.86	1.56 *	1.48	1.26
Educational status (X4)	0.99	0.99	0.63 *	0.56 ***
Expectations of long-term health care from grown children (X6)	0.93	1.20	0.99	2.40 ***
Educational status of grown children (X7)	1.06	0.73	0.57 *	0.85
Physical health of grown children (X10)	1.21 *	1.01	0.99	1.31 ***

Note: *, *p* < 0.05; **, *p* < 0.01; ***, *p* < 0.001.

**Table 6 healthcare-09-00009-t006:** The relationship between macro and micro indicators.

Criteria/Indicators	CHARLS Data from Academic Civil National Investigation (Factors from Micro-Perspective)	Weighs (*β*)	Regional Data from National Statistical Database (Factors from Macro-Perspective)	Weighs *(w)*
Criterion I	Age (X2)	0.44 (*β*2)	Regional aging level (F1)	0.44 *(w*1*)*
Criterion II	Residence (X3)	0.45 (*β*3)	Regional urbanization level (F3)	0.45 *(w*2*)*
Criterion III	Educational status (X4, X7)	0.41 (*β*4), 0.69 (*β*7)	Regional educational level (F2)	0.41 × 1/3 + 0.69 × 2/3 *(w*3*)*
Criterion IV	Expectation for the long-term health care (X6)	0.53 (*β*6)	Regional level of family care (F4)	0.53 *(w*4*)*

**Table 7 healthcare-09-00009-t007:** Four regional divisions according to the cross-section trend in 2005, 2010 and 2015.

Four Types of Region	Corresponding Provinces in China
Region 1: HA score goes up first from 2005 to 2010 and then down from 2010 to 2015	Beijing, Heilongjiang, Shaanx and Zhejiang province
region 2: HA score has been falling from 2005 to 2015	Tianjin, Shanghai, Qinghai, Jilin, Inner Mongolia, Liaoning, Hebei and Shandong province
region 3: HA score goes down first from 2005 to 2010 and then up from 2010 to 2015	Guangdong, Ningxia, Hainan, Xinjiang, Shanxi, Hubei, Gansu, Henan, Yunnan, Guangxi, Chongqing and Guizhou province
region 4: HA score keeps going up from 2005 to 2010	Hunan, Fujian, Jiangsu, Jiangxi, Anhui and Sichuan province

**Table 8 healthcare-09-00009-t008:** Regional HA index of eight economic zones of China.

Eight Economic Zones of China	HA Index in 2005	HA Index in 2010	HA Index in 2015	Average HA Index from 2005–2015
Beijing-Tianjin-Hebei	114.60	114.11	108.75	112.49
Five North China Provinces Including Beijing, Tianjin and Hebei	110.56	109.80	105.99	108.79
Three Northeast Provinces	109.37	101.33	97.90	102.87
Yangtze River Delta Region	106.10	105.51	101.70	104.43
East China Coastal Area	94.54	96.77	98.25	96.52
Central Plains Region	95.56	96.01	96.33	95.97
Pearl River Delta Region	104.36	102.86	106.45	104.56
Southwest China	88.39	88.13	92.27	89.60
Five Northwest Provinces	104.06	102.05	102.83	102.98

## Data Availability

Publicly available datasets were analyzed in this study. This data can be found here: http://charls.pku.edu.cn// 1 September 2020.

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
