# Peer review of "Data Analysis Model Design of Health Service Monitoring System for China’s Elderly Population: The Proposal of the F-W Model Based on the Collaborative Governance Theory of Healthy Aging"

_healthcare, 2020, doi:10.3390/healthcare9010009_

Round 1

Reviewer 1 Report

Comments/suggestions

Acronyms need to be explained the first time they appear

E.g. Line 18 – HA needs to be explained

-The issue of Big Data needs to be more explained and contextually interpreted. More insights about the role played by Big Data in the Chinese context and health care management.

-The notion of cooperative governance could be explicitly defined and contextually situated.

-The descriptive statistic section could be more refined and contextually situated. The HA is included in the data, even if the notion apparently is a priori known by the reader. Each domain/dimension need to be refined according to the measurement techniques (in fact, a synthesis is given in the table A1. The term CREEN must be explained, even if is a technical issue related on variable data).

The physical health (Y1) was measured in term of…

The daily activity (Y2) was defined …

Section regression logistic

-The authors need to explain why is necessary to have in the mirror the variables resulted in the first step regression and the last step regression? Which is the power of the prediction of the model and the robustness?

-The authors need to give a more interpretive perspective of the indicators retained in their analysis. Healthy ageing is a multidimensional construct, even if the models proposed are therefore abstractly defined.

- Lines 160-163 - What kind of weights is related here, which is the logistic coefficient, and what are the factors? How is calculated the scoring index and what are the assumptions?

Line 163 – what exactly means the ratio of regional education - time?

-What are the implications of HA and what are the limitations of the models? The novelty of the research could be more emphasised.

-What is the relationship between the present study and other previous similar studies?

Discussion and conclusion section might be improved.

Author Response

Dear expert,

    Thank you !

    Thank you very much!

    It is absolutely our honor to receive your review and valuable comments!

    Thank you for your constructive opinions, very helpful for us to improve our research!

    All modified text parts are highlighted in the revision paper.

    Thank you again!

    Thank you for your approval and support !

    Time is limited and may not reach the acme of perfection.

    If there is anything that still i can do, please don’t hesitate to let me know!

    We wish you a happy Christmas in advance!

    Wish you all the health and happiness forever!!

Yours sincerely,

 Wang yuhui

 wangyuhui2018@tju.edu.cn

Reviewer 2 Report

This is an interesting and important paper about healthy ageing in the Chinese population, using health survey data. Based on the data, the paper proposes a factor and weights model (F-W) model to predict healthy ageing. this could be useful for health planners.

I have some questions: 

  1. Table 2 shows the proportion of healthy ageing (HA) in four dimensions, but the overall proportion of healthy ageing was much less than the proportion in any of the separate dimensions. It was not clear to me how the dimensions were combined to create the initial HA proportion (later used for the regression). 
  2. I understand that the regressions should have eliminated multicollinearity, but I wondered if there was nevertheless any relationship between education of the older person (literate vs not) and education of their children (literate vs not) as both appear to predict HA.
  3. IN section 2.2 the authors claim: "Higher proportion of HA means lower risk of health care and lower cost of health service monitoring". Is there a reference for this?
  4. The discussion claims that the low average income of older people in China is responsible for the proportion who are in the category of HA. While this maybe true, I do not see that proven in the data presented. A reference linking income and some of the factors shown to influence HA would be helpful. If not available at a personal level from the data, perhaps some relationship between average income at the regional level might be indicative, and the discussion about this might be linked to the statement about income. 
  5. the English is understandable but needs improving in some places eg the use of articles such as "the". There are also some infelicities which would be difficult for a non native speaker to identify such as "empty elderly nesters" which would be better as "elderly empty nesters". "Empty nester" is a complex noun phrase which is best not split.
  6. Overall this is an important study of ageing in China with the potential for fruitful application in future surveys and also the power to influence policy. 

Author Response

(The authors gave the same response as above.)

Round 2

Reviewer 1 Report

please find attached file

Author Response

Dear reviewer,

    Thank you !

    Many many thanks!

    Without your help, we couldn't have done better.

    Thank you for your support and valuable suggestions, very very helpful for us to improve our research!

    We totally and successfully revised the article one by one.

    All modified text parts are highlighted in the revision paper !

    Thank you,thank you again !

    If there is anything that still i can do, please don’t hesitate to let me know !

    We wish you peace on Christmas Eve and a happy Christmas in advance !

    Wish you all the health and happiness forever !!

Yours sincerely,

Wang yuhui

Wangyuhui2018@tju.edu.cn
